# A map of directional genetic interactions in a metazoan cell

**Bernd Fischer[1†], Thomas Sandmann[2,3†], Thomas Horn[2,3†], Maximilian Billmann[2,3†], Varun Chaudhary[2,3], Wolfgang Huber[1\*], Michael Boutros[2,3\*]**

[1]Genome Biology Unit, European Molecular Biology Laboratory, Heidelberg, Germany; [2]Division of Signaling and Functional Genomics, German Cancer Research Center (DKFZ), Heidelberg, Germany; [3]Department of Cell and Molecular Biology, Faculty of Medicine Mannheim, Heidelberg University, Heidelberg, Germany

**Abstract** Gene–gene interactions shape complex phenotypes and modify the effects of mutations during development and disease. The effects of statistical gene–gene interactions on phenotypes have been used to assign genes to functional modules. However, directional, epistatic interactions, which reflect regulatory relationships between genes, have been challenging to map at large-scale. Here, we used combinatorial RNA interference and automated single-cell phenotyping to generate a large genetic interaction map for 21 phenotypic features of *Drosophila* cells. We devised a method that combines genetic interactions on multiple phenotypes to reveal directional relationships. This network reconstructed the sequence of protein activities in mitosis. Moreover, it revealed that the Ras pathway interacts with the SWI/SNF chromatin-remodelling complex, an interaction that we show is conserved in human cancer cells. Our study presents a powerful approach for reconstructing directional regulatory networks and provides a resource for the interpretation of functional consequences of genetic alterations.

**\*For correspondence:** whuber@embl.de (WH); m.boutros@dkfz.de (MB)

†These authors contributed equally to this work

**Competing interests:** The authors declare that no competing interests exist.

**Reviewing editor**: Christopher Glass, University of California, San Diego, United States

## Introduction

Genes display epistatic (genetic) interactions, whereby the presence of one genetic variant can mask, alleviate or amplify the phenotypic effect of other variants (*Bateson, 1907*). Genetic interactions have profound effects during normal development and in disease (*Moore, 2003*; *Ashworth et al., 2011*).

Two genes show a genetic interaction if the phenotype of allele combinations (or double perturbations) is different from the expected combination of the individual phenotypes; this statistical definition of genetic interactions was introduced by *Fisher (1918)*. Mathematically, Fisher defined a genetic interaction as an interaction term between genetic covariates in a (statistical) linear model for a phenotype. Genetic interactions occur, for example, when a cellular process is controlled by two parallel pathways: loss of gene products in one pathway are buffered, and only when the both pathways are disturbed, a phenotype is observed. Conversely, the phenotype of closely collaborating proteins, for example, in a protein complex, is often not further enhanced when more than one of its gene products are depleted (*Dixon et al., 2009*). Systematic screens for genetic interactions have been performed in yeast (*Tong et al., 2004*; *Schuldiner et al., 2005*; *Costanzo et al., 2010*; *Ryan et al., 2012*), *Escherichia coli* (*Nichols et al., 2011*; *Babu et al., 2014*), *Caenorhabditis elegans* (*Lehner et al., 2006*) and metazoan cells (*Bakal et al., 2008*; *Horn et al., 2011*; *Bassik et al., 2013*; *Laufer et al., 2013*; *Roguev et al., 2013*). These approaches have successfully unravelled symmetric relationships, such as pathway and complex co-membership (*Baryshnikova et al., 2013*). However, the observation of a statistical genetic interaction implies no directionality between the genes: it does not take into account a possible order of relationships, for example, their temporal relation in a signalling cascade. Directionality is present if one gene product positively or negatively regulates the

**eLife digest** Genes encode instructions that control our physical characteristics, known as traits. Although some traits are controlled by the activity of a single gene, most traits are influenced by the activities of multiple genes.

The genes that influence a particular trait may work independently of each other. However, it is also possible for the genes to interact so that one gene may mask or amplify the effect of another gene. Although gene interactions were first described almost 100 years ago, it has been difficult to identify them and work out the direction of these interactions (i.e., does gene A affect gene B, or vice versa?).

Fischer, Sandmann et al. have now studied the interactions between the genes involved in 21 different traits of fruit fly cells. A technique called RNA interference was used to lower the expression of the genes in different combinations, which made it possible to analyze any changes in the traits that occurred when particular genes were not working properly. Fischer, Sandmann et al. took hundreds of thousands images of the cells and analyzed the changes in cell shape, cell size, cell division and other traits. Next, they developed a method to infer the directions of the interactions between individual pairs of genes from the data and then made a map of the genetic interactions for the traits.

This map was able to reconstruct the known order of activity of genes during cell division and other cell processes. Furthermore, it revealed previously unknown interactions between genes. For example, genes involved in the Ras signaling pathway—which promotes cell growth and is frequently mutated in human tumors—interacted with genes that encode a group of proteins called the SWI/SNF complex. This complex alters how DNA is packaged in cells to control the expression of genes, and these gene interactions may play an important role in the control of cell growth by Ras signaling.

The approach developed by Fischer, Sandmann et al. can shed light on the interactions between genes that produce complex traits of cells. In future, this approach might be helpful to find out which genetic differences between individuals alter the effectiveness of drug treatments, and the impact of using combinations of drugs to treat diseases.

activity of the other, if its function temporally precedes that of the other, or if its function is a necessary requirement for the action of the other. Such directional genetic interactions were described by *Bateson (1907)*, and observing epistatic interactions between genes has been a powerful method to organise them into functional pathways (*Phillips, 2008*). In special cases, for example, in which one of the two interrogated genes has no phenotype itself, it has been possible to infer directionality by comparing the gene–gene phenotype to the single gene phenotype (*Drees et al., 2005*; *St Onge et al., 2007*). However, in many cases genes that show genetic interactions have phenotypes themselves, and more general methods to predict directionalities have been lacking.

We report the largest map of multi-phenotype genetic interaction profiles in metazoan cells to date. Our map comprises 1367 *Drosophila* genes implicated in cellular processes including signalling, chromatin and cell cycle regulation. We quantitatively scored 21 phenotypes in cultured cells using automated imaging. First, we inferred gene functions based on statistical genetic interactions, predicted protein complexes and clustered processes by similarity. Moreover, in a novel analytical approach, we employ the multivariate nature of the phenotypes to compute directional genetic interactions, and we show that they reveal the logical and temporal dependencies between functional modules. For example, we were able to reconstruct the temporal order in which protein complexes are active during mitosis. Furthermore, we demonstrate an epistatic relationship between Ras/MAPK signalling and SWI/SNF regulators of chromatin remodelling, which we confirmed in vivo. We demonstrate that directional genetic interactions can be identified at a large scale and mapped to other genomic data sets to identify regulatory relationships.

## Results

### Quantitative genetic interactions by high-throughput imaging

We generated the largest map of multi-phenotype genetic interaction profiles in metazoan cells to date by co-depleting gene pairs by RNAi in cultured *Drosophila* S2 cells, high-throughput imaging of

single-cell phenotypes, and modelling of gene–gene interactions (*Figure 1A*). We selected 1367 genes implicated in key biological processes, that is, signalling, chromatin biology, cell cycle regulation and protein turnover control (*Supplementary file 1*). Each of these 1367 target genes was tested against 72 query genes in all pairwise knockdown combinations (2 × 2 dsRNAs), following previously established approaches (*Casey et al., 2008*; *Horn et al., 2011*; *Laufer et al., 2013*) (*Figure 1—figure supplement 1*). The 72 query genes were selected from an initial single-gene screen on the 1367 genes, to cover a range of phenotypes, processes and protein complexes (*Figure 1—figure supplement 2* and *Supplementary file 2*). After 5 days, cells were fixed and stained for DNA, α-tubulin, and Ser9-phosphorylated histone 3, a mitosis marker. Cells were imaged by automated whole-well fluorescence microscopy, and phenotypic features were extracted using an image analysis pipeline (see 'Materials and methods'). On average, 15,962 cells were imaged and analysed per well. Single-cell measurements were aggregated into 328 cell population features per experiment such as cell number, mitotic index, nuclear and cellular area, and other descriptors of shape and morphology (*Supplementary file 3*). 162 features were highly reproducible between replicates, with Pearson correlation >0.6 (*Figure 1B–C*). Using a step-wise feature selection algorithm, we determined a subset of 21 features (*Supplementary file 4*) that non-redundantly captured the range of phenotypes. The algorithm ensured that each feature contained new information not yet covered by the already selected features (*Figure 1D* and 'Materials and methods') (*Laufer et al., 2013*).

Of the 1367 single gene knockdowns, 499 genes showed a statistically significant phenotype in at least one feature (false discovery rate, FDR 1%). Collectively, the phenotypes provided a broad overview of biological processes that impact cell proliferation and growth, cell shape and cell cycle regulation. For example, depletion of *ida*, a subunit of the anaphase-promoting complex (APC/C), led to an increased mitotic index as previously described (*Bentley et al., 2002*) (*Figure 1E*). In contrast, cell number and mitotic index decreased upon depletion of *stg*, the *Drosophila* CDC25 ortholog required for mitotic progression, which resulted in enlarged cells with large nuclei (*Edgar and O'Farrell, 1990*; *Cui and Doe, 1995*) (*Figure 1E*). Knockdown of *Arpc1*, a core member of the actin-related protein Arp2/3 complex, led to elongated, spindle-like cell morphologies detected by an increase in cell eccentricity (*Machesky et al., 1994*) (*Figure 1E*). Thus, the measured phenotypes reflect known biological functions.

All 98,424 pairwise knockdown combinations of these 1367 target genes and the 72 query genes were tested four times each (2 × 2 dsRNAs). We filtered out genes with suboptimal dsRNAs (low efficiency or specificity) by assessing the phenotypic concordance between the two dsRNAs targeting different regions of the same gene (referred to as sequence-independent dsRNAs). For more than 94% (1293 of 1367) of genes, phenotypic profiles were highly correlated (*Figure 1F*, *Supplementary file 5*), and these genes were used in further analysis. Genetic interactions were calculated using a multiplicative model (*Horn et al., 2011*). 12361 gene pairs (13%) showed significant statistical genetic interactions in at least one phenotype (FDR 1%, moderated *t*-test). Compared to an analysis of the cell number phenotype alone, which detected 2357 genetic interactions, the number of observed genetic interactions was more than fivefold higher in the multi-phenotype analysis (*Figure 2A*). Interactions derived from different phenotypes were non-redundant, for example, genetic interactions affecting the number of cells displayed only weak correlation with those affecting mitotic index (correlation coefficient: 0.33) or cell diameter (correlation coefficient: −0.32) (*Figure 2B*). For example, depletion of *Rho1* mRNA alone interfered with cytokinesis, resulting in large, multinucleated cells and reduced cell number (*Figure 2C*). In contrast, RNAi against *Dlic*, the dynein light intermediate chain, led to a moderate reduction in cell number but did not affect nuclear size. As predicted from the single-RNAi phenotypes, combined targeting of *Rho1* and *Dlic* reduced cell numbers further, with no evidence of a genetic interaction for this phenotypic feature. In contrast, their co-depletion resulted in a negative genetic interaction on cell and nuclear sizes, indicating that *Dlic* is epistatic to *Rho1*. Similarly, co-depleting *Not3* (a member of the CCR4-NOT deadenylase complex) and the growth regulator *hippo* (*hpo*) did not reveal a significant genetic interaction affecting total cell number. Yet, the same RNAi combination reversed the mild decrease in mitotic index observed upon RNAi of each single gene, and increased the fraction of mitotic nuclei (*Figure 2D*). Thus, genetic interactions affect multiple phenotypes, with each phenotype contributing independent information.

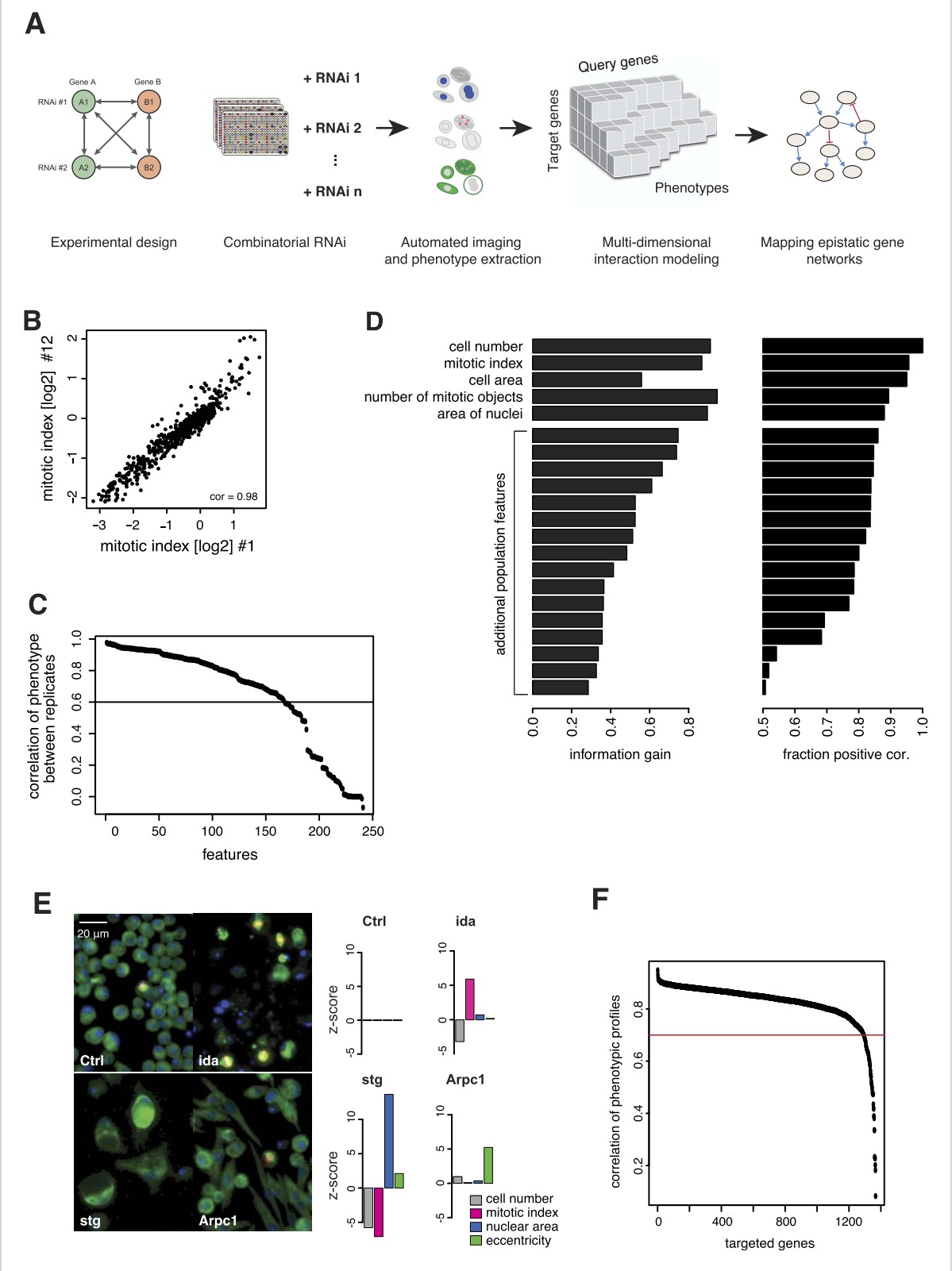

**Figure 1**. Combinatorial RNAi to map multi-phenotype genetic interactions. (**A**) Workflow for multi-phenotype genetic interaction analysis by RNAi. (**B**) Reproducibility of phenotypic measurements. Plot shows replicate measurements for 1293 target genes at the beginning and end of the screening campaign. No batch effects on phenotypes were observed. (**C**) Each point in the plot corresponds to one of the phenotypic features. The y-axis shows the Pearson correlation coefficient of the feature's values between two replicates. Along the x-axis, features are ordered by their correlation coefficient.
*Figure 1. continued on next page*

*Figure 1. Continued*

(**D**) Selection of non-redundant features proceeded step-wise, starting with cell number, mitotic index and cell area. In the left panel, the x-axis shows the information gain (as measured by the correlation of the residuals between replicates) for the selected features. Features are ordered as selected. In the right panel, the x-axis shows the fraction of positively correlated residual features remaining, which is used as a stop criterion (*Laufer et al., 2013*). (**E**) Representative image regions are shown for negative control (Ctrl), *imaginal discs arrested* (*ida*), *string* (*stg*) and *actin-related protein 2/3 complex, subunit 1* (Arpc1). Bar charts display measured quantitative features. (**F**) Two independent dsRNA reagents per gene were used to assess on-target specificity. The plot shows the correlation coefficient (r) between the two reagents across all phenotypic features and 72 query dsRNAs. Only genes with r > 0.7 (red line) were included in further analyses.

The following figure supplements are available for figure 1:

**Figure supplement 1**. Experimental design.

**Figure supplement 2**. Selection of query genes.

**Figure supplement 3**. Comparison of phenotypes.

## The landscape of statistical genetic interactions in a metazoan cell

To obtain an initial overview of the patterns of genetic interactions, we clustered the interaction profiles of each of the 1293 genes across the 21 phenotypes (*Figure 3—figure supplement 1* shows nine representative phenotypic features). We constructed a correlation graph that represents the similarity between interaction profiles across all phenotypes for each gene pair (*Figure 3A*). The information generated by the multi-phenotyping dissected numerous biological processes in detail. For example, DNA-dependent RNA Pol II core components clustered together and were connected to general transcription factors, mediator and SAGA components, which are known to regulate Pol II transcriptional activity. Also, the SWI/SNF chromatin remodelling complex members *Bap60*, *brm*, *mor*, *osa*, *Snr1* clustered together, representing the *osa*-containing complex state (*Moshkin et al., 2012*).

Tor signalling (i.e., *Tor*, *Rheb*, *S6k*, *dm*, *Pdk1*) was connected to a cluster containing the translational apparatus, as expected (*Figure 3B*). Beside genes belonging to the RNA Pol I and Pol III complexes, the translational apparatus cluster contained putative RNA helicases (*CG32344*, *CG9630*, *kz*, *pit* and *Rs1*) and the RNA-binding methyltransferase *CG8545*, suggesting a function related to ribosome biogenesis. Interestingly, *Dbp45A*, a putative ATP-dependent RNA helicase of the DEAD box protein family, was linked to several genes involved in ribosomal biogenesis as well as with genes involved in DNA damage repair (*Figure 3B*), suggesting that it might facilitate communication between the translation machinery and DNA repair and apoptosis.

The map exposed two tight and well-separated clusters of genes that have functions in DNA damage sensing and repair, and in the regulation and execution of apoptosis (*Figure 3B*). Intriguingly, the cluster containing *p53*, *Ice*, *CG7922*, *Ikb1* also contained genes known to regulate the cytoskeleton and vesicle trafficking. This cluster also showed similarities of genetic interaction profiles with *hpo* and *14-3-3ε*, two core members of the Hippo signalling pathway.

In the genetic interaction map, all γ-tubulin ring complex (γTuRC) members mapped in a dense cluster surrounded by the motor proteins of the dynein/dynactin complexes as well as auxiliary regulators of the mitotic spindle (*Figure 3B*). Also in close proximity to the mitotic spindle we found the APC/C, which drives the metaphase-to-anaphase transition by priming several cell cycle regulators for ubiquitin-mediated degradation by the proteasome. Interestingly, the dynactin member *Arp10* was found to have an 'APC/C-like' interaction profile, suggesting a role in supporting APC/C function. Moreover, our map showed that the APC/C is functionally linked to the proteasome via *Cdc27*, the motor protein *Klp61F*, the APC/C activator *fzy* (CDC20) and the ubiquitin-conjugating enzyme *vih*. We observed that gene pairs with high correlation coefficients are enriched for experimentally validated protein–protein interactions (*Figure 3—figure supplement 2*) (*Guruharsha et al., 2011*). Overall, the different protein complexes and functional modules were well separated in the genetic interaction map.

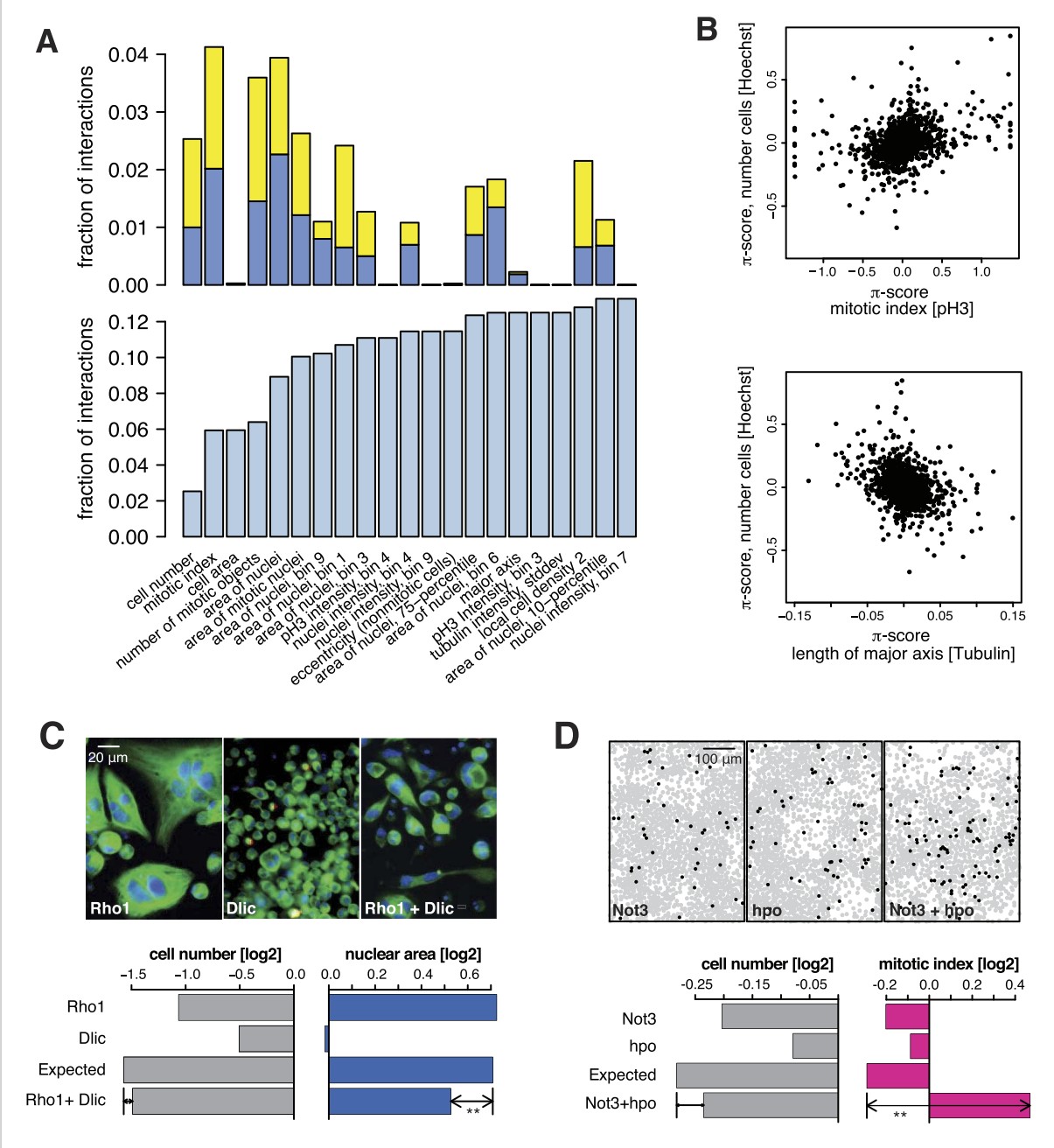

**Figure 2**. Genetic interactions across multiple phenotypes. (**A**) On top, the fraction of genetic interactions over all gene pairs (at FDR 0.01) for each feature is shown. Blue: negative interactions, yellow: positive interactions. The lower panel shows the cumulative distribution of genetic interactions. The k[th] bar shows the fraction of gene pairs that show a genetic interaction in at least one of the first k features. (**B**) Genetic interactions for different phenotypes were non-redundant. Plots compare genetic interactions affecting cell number, mitotic index and length of major cell axis. (**C** and **D**) Many genetic interactions were phenotype-specific. Interactions at a FDR of 0.01 for a moderated *t*-test (See 'Materials and methods') are marked by (**). Images in (**C**) show cells after RNAi of *Rho1* (left), *Dlic* (center) and both together (right). Bar charts show single-gene effects, effects expected without a genetic interaction, and effects observed by the double-RNAi. Images in (**D**) show cell centres identified by image analysis after RNAi of *Not3* (left), *hpo* (center) and both (right). Black and grey colours indicate whether a cell was pH3-positive or negative (mitotic state).

## Inferring directional genetic interactions

While symmetric (non-directional) genetic interactions reflect a collaborative role of genes, for instance, between members of a protein complex, this approach does not provide an ordering of the

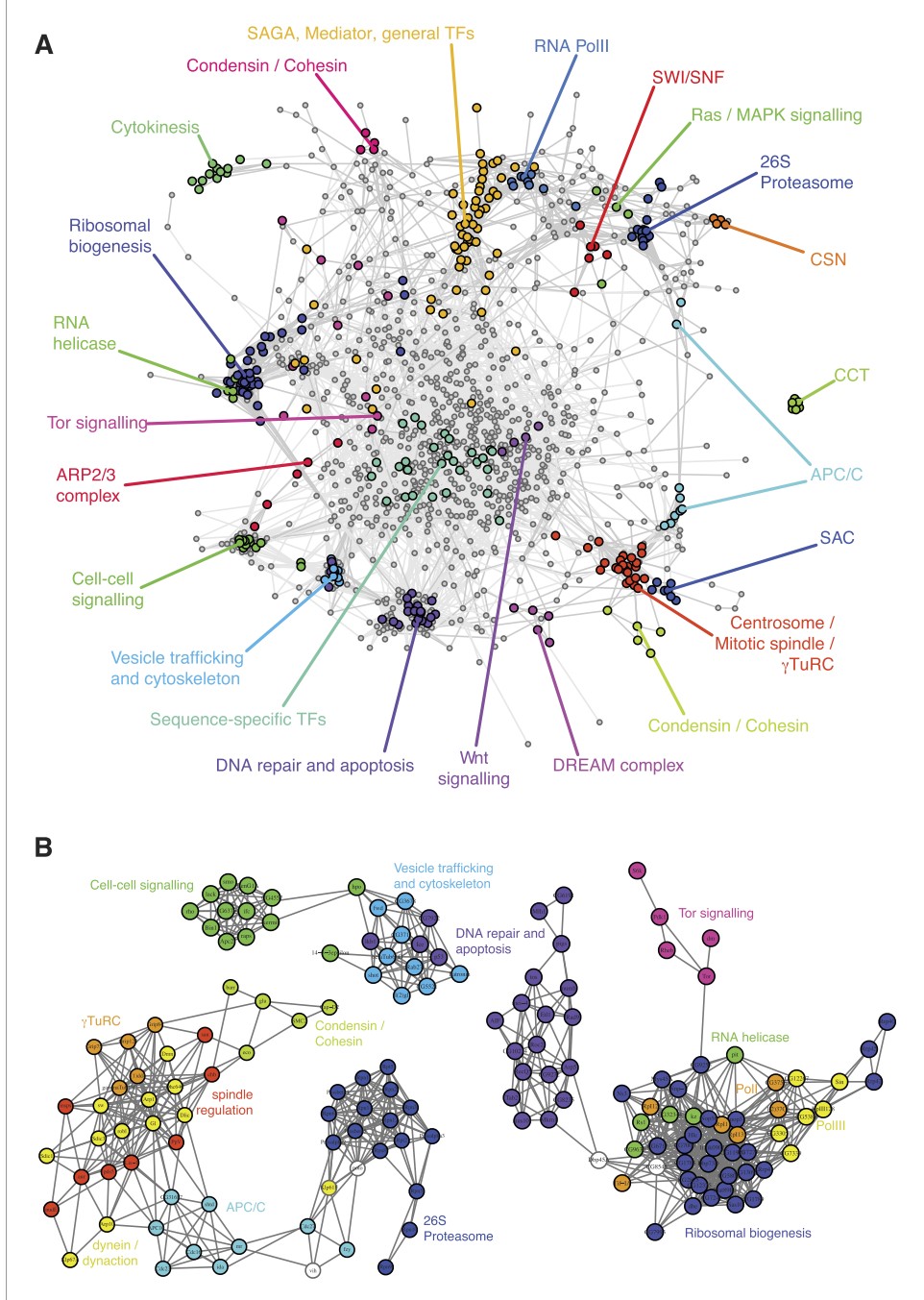

**Figure 3**. A landscape of statistical genetic interactions. (**A**) Correlation network of genetic interaction profiles. Pearson correlation coefficients were computed for each pair of genes using the profile of all genetic interactions of these two genes to all other genes in all phenotypic features. An edge is drawn in the graph for each gene pair with correlation coefficient ≥0.6. Genes were placed by a graph layout algorithm (Fruchtermann–Reingold). Genes with similar genetic interaction profiles are proximal. The colour code depicts different biological processes that were separated by the correlation network. (**B**) Subgraphs of the correlation graph shown in *Figure 2* highlight the wiring within and between the annotated complexes.

The following figure supplements are available for figure 3:

**Figure supplement 1**. Heatmap of multi-parametric genetic interactions.

**Figure supplement 2**. Genetic interaction profiles and protein complexes.

proteins according to their temporal or regulatory activities. We leveraged the multivariate nature of our data to identify the direction of genetic interactions by testing each pair of genes (A and B) for one of two directional relationship models: A modifies the phenotype of B, or B modifies the phenotype of A. Moreover, we assigned a sign to the relationship, which indicates whether the modification is of an alleviating or aggravating nature.

For each gene pair, we analysed the multivariate phenotypes of both single as well as double perturbations. This is exemplarily shown in *Figure 4A*. The first column depicts multivariate single (in blue and orange) and double knockdown (in grey) phenotypes for five different experiments. In a first step, we computed the multivariate genetic interaction scores (π) as described above (*Figure 4A*, second column). A geometric interpretation to visualize the relationship of the genetic interactions to the single gene phenotypes is shown in the third column of *Figure 4A*. We explain the five exemplary cases below.

First, consider two genes A and B with distinct multivariate phenotypes. In *Figure 4A*, this situation is depicted by the blue and orange arrows that represent the effects of the single-gene knockdowns. When A and B do not genetically interact, the phenotype of the double-knockdown (grey arrow; AB) is simply the combination (vector sum) of the two individual arrows (*Figure 4A*). We call this the non-interacting model (NI). Second, when A and B genetically interact, the multivariate phenotype of the double knockdown (AB) is different from the vector sum (NI). This difference, geometrically depicted by the black arrows, defines the genetic interaction (π). If the double knockdown phenotype is an amplified or dampened version of one of the single gene knockdown phenotypes, then the genetic interaction (black arrow) is parallel or antiparallel to the phenotype of either one of the single knockdowns (blue or orange arrow: *Figure 4A*). Together, this geometric information indicates the directionality of the interaction (i.e., which gene modifies which), and whether it is alleviating (antiparallel) or aggravating (parallel). For example, if the double knockdown phenotype of gene A and gene B is similar to the single knockdown of gene A, we call this an alleviating interaction from A to B (A → B). This indicates that a functional gene A is required for the phenotype of gene B (i.e., gene A is functionally upstream of gene B), while the absence or presence of B has a limited effect on the phenotype of A. Similarly, *Figure 4A* shows an example of an aggravating interaction from gene A to gene B (A ⊣ B), where loss of gene A's function leads to an even stronger phenotype of gene B, while the phenotype of gene A is unchanged. The two remaining cases, alleviating and aggravating directional interactions from gene B to gene A are shown in *Figure 4A* (B → A, B ⊣ A). Alternatively, if the double knockdown phenotype (AB) is not merely increasing or decreasing the amplitude of one of the single gene phenotypes, no direction is assigned (*Figure 4—figure supplement 1*).

We systematically determined whether such an epistatic relationship could be identified for each of the 93,096 gene pairs by testing if the 21-dimensional genetic interaction vectors were (anti-) parallel to either of the two single knockdown vectors ('Materials and methods', *Figure 4—figure supplement 2*). Using a conservative threshold, we found directional epistatic interactions between 1344 pairs of genes. Of these, 1040 were alleviating and 304 aggravating (*Supplementary file 6*). This directional genetic interaction network provided a rich snapshot of regulatory, causal and temporal relationships. For instance, we detected an alleviating interaction between the APC/C member *Cdc23* and the cytokinesis regulator *sti*. After knockdown of *Cdc23*, we recorded reduced growth and slightly smaller nuclear area (*Figure 4B–C*, blue arrows for the two independent dsRNAs). Knockdown of *sti* also reduced growth, but in addition led to a strong increase in nuclear area (*Figure 4B–C*, orange arrows). The double knockdown of *Cdc23* and *sti*, however, almost completely reproduced the phenotype of *Cdc23* alone (*Figure 4C*, black arrows for 2 × 2 dsRNA design). This directional alleviating interaction between the genes reflects the known, consecutive functions of the encoded proteins. Without *Cdc23*, cells arrest in metaphase, leading to the observed reduction in cell number (*Figure 4B*, top). Without *sti*, a component of the contractile ring, which executes the abscission of the cytoplasm to form the two daughter cells (*D'Avino et al., 2004*), cells undergo karyokinesis (division of the nucleus) and re-replicate their genome without being able to divide their cytoplasm, resulting in giant cells with large nuclei and often multi-nucleated cells (*Figure 4B*, middle). However, when *Cdc23* is depleted together with *sti*, the increase of nucleus size is avoided (*Figure 4B*, bottom) since depletion of *Cdc23* arrests cells in metaphase thereby preventing re-replication (*Zielke et al., 2008*), explaining the detected directional interaction (*Figure 4D*).

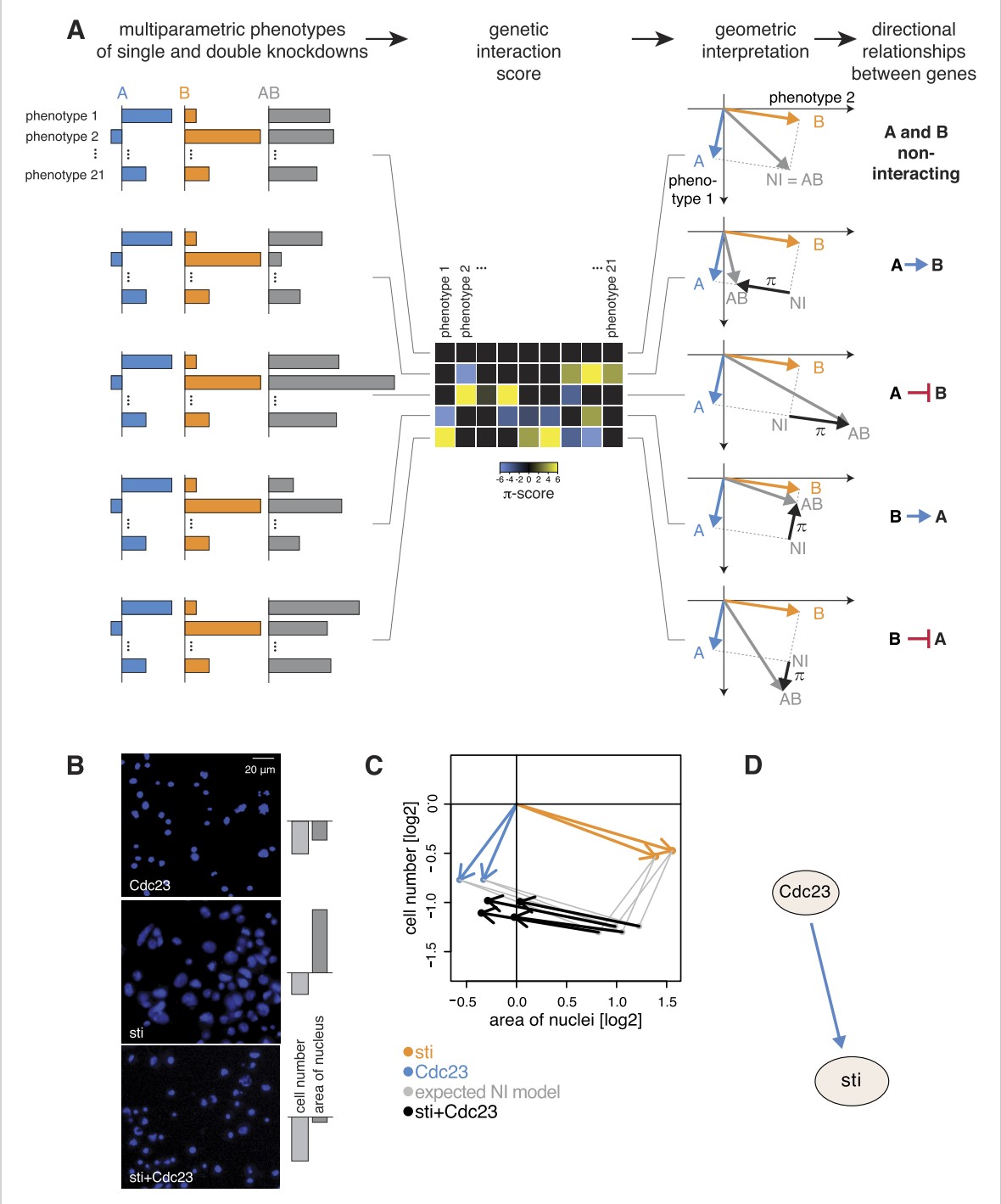

**Figure 4**. Deriving directional genetic interactions. (**A**) Multiparametric phenotypes are extracted for single and double knockdowns. Genetic interaction scores are computed for each double knockdown experiment. The schematic plots in the third column show the model for identifying directional genetic interactions between gene A and gene B using two exemplary phenotypes. The single knockdown phenotypes of genes A and B and the measured double knockdown phenotypes (AB) are depicted as arrows. The expected double knockdown phenotype for non-interacting (NI) genes, which is the sum of the single gene effects, is depicted by the symbol NI. Black arrows depict the genetic interaction π. The first row shows the case where genes A and B are not interacting. Below, four types of interaction between the genes A and B are shown: gene A is alleviating to gene B, gene A is aggravating to gene B; and in reverse, B alleviates or aggravates gene A. Whenever the genetic interaction (black arrows) is parallel or anti- parallel to one of the single gene effects, a directional genetic interaction is called. (**B–D**) A directional interaction detected between *Cdc23* and *sti*. (**C**) The two orange and two blue arrows show the phenotypes (nuclei area and cell number) of the two dsRNAs designed for *sti* and *Cdc23*. The grey dots show the expected double knockdown effect for the two genes. The black arrows, indicating the genetic interaction, are directed opposite to the phenotype of *sti*, indicating that functional

*Figure 4. continued on next page*

*Figure 4. Continued*

*Cdc23* is required for the phenotype of *sti*. (**D**) Graphical annotation of the detected alleviating epistasis from Cdc23 to *sti* as shown in (**B–C**).

The following figure supplements are available for figure 4:

**Figure supplement 1**. Directed genetic interactions.

**Figure supplement 2**. Inference of directed genetic interaction of Cdc23 and sti.

## An epistatic network of mitotic genes

Next, we studied the directional genetic interactions of key regulators of the cell cycle (*Figure 5A*). The mitosis master regulator *polo* was observed as alleviating epistatic to components that are required for the initial mitotic phases, including the γTuRC, which serves as the structural basis of the mitotic spindle. In contrast, the ubiquitin-conjugating enzyme *vih* showed directional interactions to components that are active at later stages. Depletion of γTuRC members aggravated the cellular phenotype of the spindle assembly checkpoint (SAC). Dynein/dynactin motor proteins mediated a chain of directional epistatic interactions between the SAC and the APC/C. Moreover, whereas SAC knockdown had an aggravating effect on the phenotype of condensin/cohesion, the latter relied on a functional APC/C. The phenotypes of cytokinesis regulators including *sti*, *pav*, *tum*, *Rho1* or *scra* were strongly dependent on APC/C as well as on dynein/dynactin motor proteins. Our data also showed that *vih*, the ubiquitin-ligase APC/C, as well as cytokinesis executers regulated the phenotype of the SCF ubiquitin-ligase core component *Skp2*. In contrast, *Elongin-B*, another ubiquitin-ligase core component, itself regulated the phenotype of the SAC, condensin/cohesin components and cytokinesis regulators. This unbiased mapping and automated inference of directional epistatic interactions, made possible by the multivariate phenotyping approach, revealed a detailed circuit diagram of regulatory, temporal and causal relationships of complexes that function during the M-phase of the cell cycle (*Figure 5B*).

## An epistatic link between Ras signalling and chromatin remodelling

Inspection of our epistatic map highlighted an unexpected link between chromatin remodelling and RAS signalling. We found genetic interactions between the SWI/SNF complex with both the SWI/SNF-associated chromatin remodeller *dalao* (*Papoulas et al., 2001*) and the Ras inhibitor *RasGap1* (*Figure 6A*). Moreover, *dalao* and *RasGap1* also both interacted with other Ras signalling pathway components. Directional epistasis was evident in several of these genetic interactions. While *dalao* alleviated the knockdown phenotype of SWI/SNF, *RasGap1* aggravated it (*Figure 6A–B*). *RasGap1* is a known negative regulator of Ras signalling (*Gaul et al., 1992*), and consistent with that, its single knockdown increased cell proliferation. We found that additional knockdown of SWI/SNF complex members (*mor*, *brm*, *Bap60*, *Snr1* or *osa*) compensated the *RasGAP1*-dependent increased proliferation (*Figure 6C*), suggesting that constitutive active Ras signalling requires SWI/SNF function. In addition, cell area and eccentricity were increased after the double knockdown compared to the single-knockdown (*Figure 6C*). Next, we tested whether mutations in SWI/SNF complex components could suppress a Ras gain-of-function rough eye phenotype in the *Drosophila* eye (*Zipursky and Rubin, 1994*). As shown in *Figure 6D*, the phenotype caused by the expression of constitutively active *Ras85D$^{V12}$* in the eye was rescued by mutant alleles of the SWI/SNF members *osa*, *Snr1* or *brm* (*Figure 6D–E*).

Next, we tested whether human KRAS-mutant cancer cells might specifically depend on the SWI/SNF complex. We investigated this by knockdown of SWI/SNF complex members in the colon cancer cell line HCT116, which harbours an activating KRAS G13D mutation (KRAS$^{G13D/+}$), and in a derived cell line in which the mutant KRAS allele was deleted (KRAS$^{KO/+}$). Whereas there was no difference in cell growth between the two isogenic cell lines (*Figure 6—figure supplement 1*), knockdown of individual SWI/SNF members SMARCA4, SMARCB1 and ARID1A caused a significant decrease in viability selectively in the context of constitutively active KRAS (*Figure 6F*). Together, this indicates that SWI/SNF complex members genetically interact with activated Ras signalling.

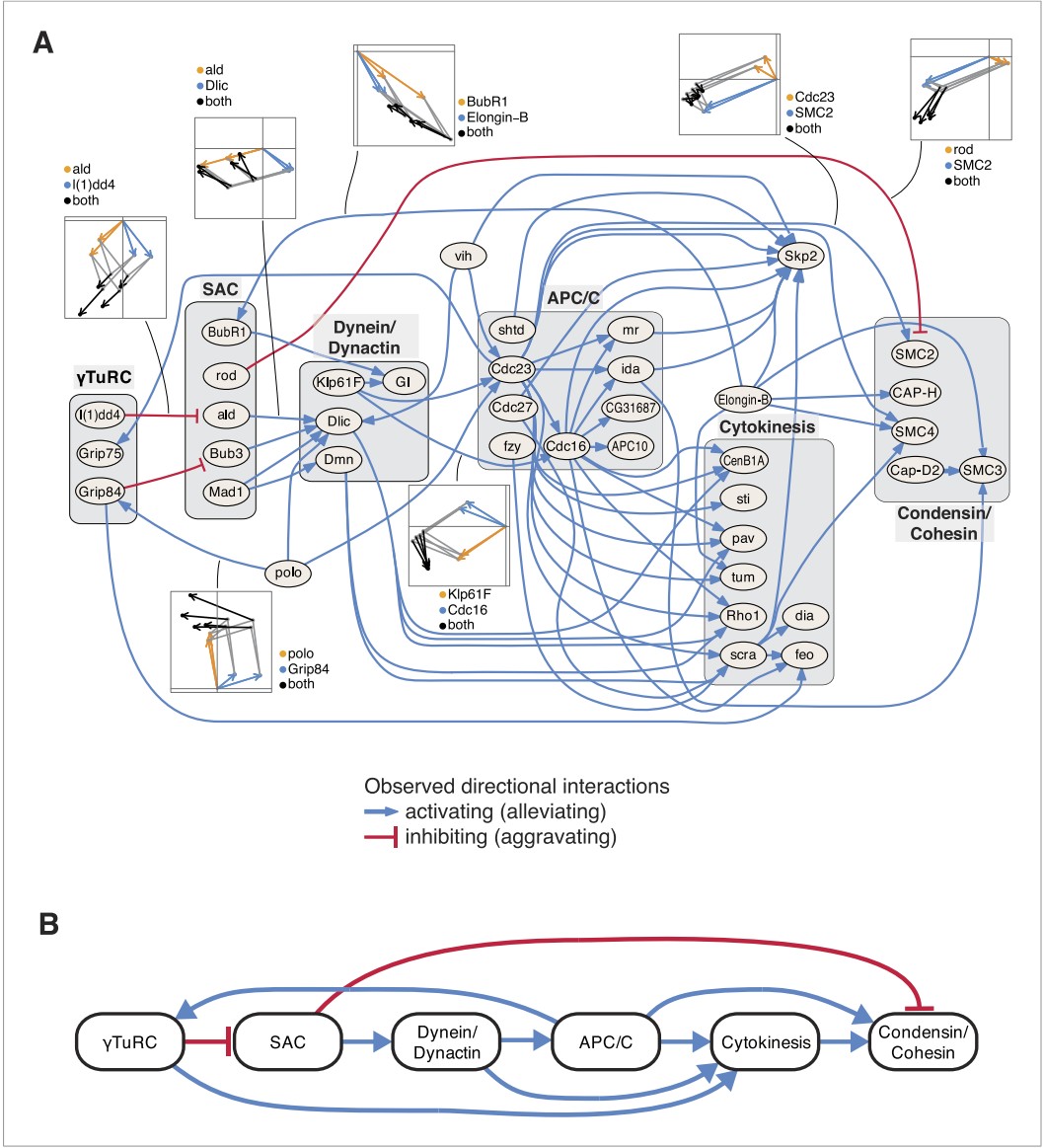

**Figure 5**. An epistatic network of mitotic genes. (**A**) Graph showing data-derived epistatic interactions of genes that are key regulators of cell cycle events, including *polo*, and members of the γTuRC, SAC, APC/C, the dynein/dynactin complex, sister chromatid cohesion complexes and cytokinesis genes. Blue arrows show alleviating epistatic interactions and red arrows show aggravating epistatic interactions. Exemplarily, the underlying data is shown for epistatic interactions. (**B**) Epistatic network of complexes that regulate mitotic events, as derived from the epistatic interactions between the members of the complexes.

## Mapping directional relationship on genes recurrently mutated in human cancer

The detection of recurrent mutations in genes and pathways in cancers (*Tamborero et al., 2013*) has resulted in a growing catalogue of known cancer mutations, however, in any individual tumour, only a subset of these mutations is present, and how they cooperatively produce the cancer phenotype is often not well understood. We aimed to use our directional interaction network to predict relationships between recurrent cancer mutations. For example, we found an alleviating directional interaction between *Pten* (PTEN) and *gig* (TSC2) (*Figure 7A*), as previously described (*Song et al., 2012*). We also identified upstream and downstream regulators: for instance, our data predict that the ARP2/3 complex member *Arp3*, a regulator of the actin cytoskeleton, aggravates the phenotype of the Ras GTP

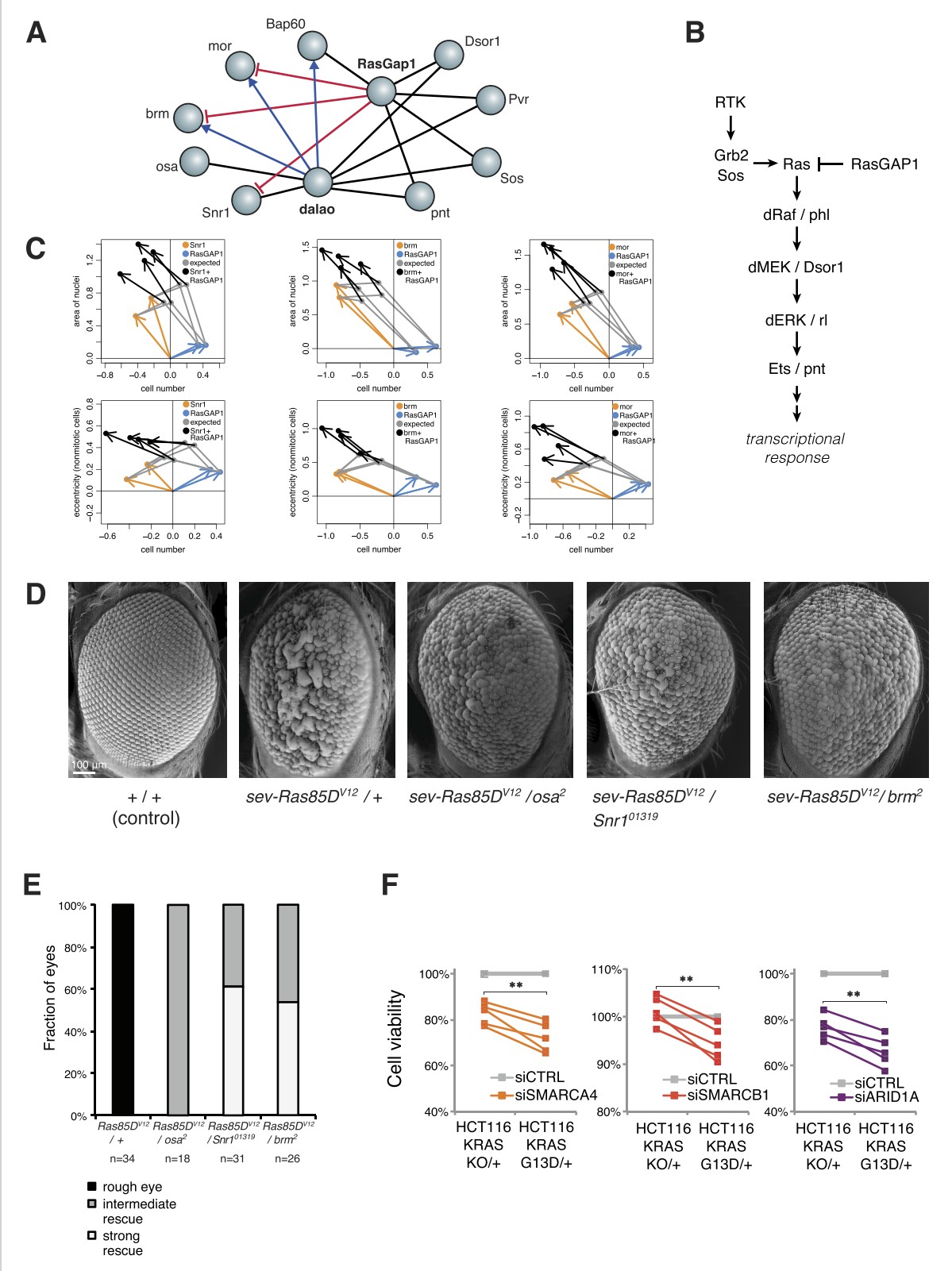

**Figure 6**. Genetic interactions between Ras signalling and chromatin remodelling. (**A**) Edges show the detected genetic interactions in the interaction screen of members of the SWI/SNF complex with *RasGAP1* and members of the Ras pathway. An edge is drawn if a significant interaction (FDR 0.01) was detected for at least one phenotype. Edges with attributed epistatic interactions are shown in blue (alleviating epistasis) and red (aggravating epistasis). Black edges indicate gene pairs for which a genetic interaction was detected, but directionality could not be assigned. (**B**) Scheme of the Ras/MAP kinase

*Figure 6. continued on next page*

**Figure 6. Continued**

pathway. *RasGAP1* inhibits the activity of the pathway by catalysing the intrinsic GTPase activity of *Ras*. (**C**) Exemplary the cell number, area of nuclei and eccentricity phenotypes are shown that guide the directionality estimate. Blue arrows indicate the RasGap1 phenotype, the SWI/SNF phenotype is depicted by orange arrows and the black arrows show the genetic interaction between SWI/SNF and RasGap1. (**D**) In vivo validation of genetic interaction of SWI/SNF and constitutively active Ras85D. From left to right: Normal eye in wild type flies (Oregon-R, +/+). Rough eye phenotype upon expression of Ras85D$^{V12}$ under the *sevenless* promotor (*sev-Ras85D$^{V12}$/+*). Strong rescue after heterozygous removal of the SWI/SNF complex members *osa* (*sev-Ras85D$^{V12}$/osa$^2$*), *Snr1* (*sev-Ras85D$^{V12}$/Snr1$^{01319}$*) and *brm* (*sev-Ras85D$^{V12}$/brm$^2$*). (**E**) Quantification of the rescue of the rough eye phenotype. Black colour: expression of Ras85D$^{V12}$ in the eye of otherwise wild-type flies results in a rough eye phenotype. Grey to white colour scale shows the strength of the rescue of the rough eye phenotype in flies expressing Ras85D$^{V12}$ in the eye and with heterozygous mutants of SWI/SNF complex members *osa* (*sev-Ras85D$^{V12}$/osa$^2$*), *Snr1* (*sev-Ras85D$^{V12}$/Snr1$^{01319}$*) and *brm* (*sev-Ras85D$^{V12}$/brm$^2$*). (**F**) Cell viability of isogenic HCT116 cells 96 hr after siRNA against members of the SWI/SNF complex (*SMARCA4*, *SMARCB1*, and *ARID1A*). Right: parental HCT116 KRAS mutant cells (KRAS$^{G13D/+}$); left: isogenic cell line HCT116 KRAS wt (KRAS$^{KO/+}$). The experiment was performed in four biological replicates.

The following figure supplement is available for figure 6:

**Figure supplement 1**. Ras-induced cell growth requires SWI/SNF complex function.

exchange factor *Sos* (**Figure 7B**). Another example is given by the epistatic interactions of *Myb* (MYB), an essential member of the DREAM complex. This complex regulates cell-cycle dependent gene expression (**Sadasivam and DeCaprio, 2013**) and is recurrently mutated in cancer. Upstream of *Myb* we detected its known regulators *mip120* and *mip130* (**Figure 7C**) (**Beall et al., 2004**). Moreover, our data indicate functions of central mitosis regulators (*polo*, *Elongin-B*, *fzy*) upstream of *Myb*, while *SMC4*, which is involved in chromosomal maintenance, and many cytokinetic genes are downstream. The DREAM subcomplex members *Myb*, *mip120*, *mip130* have been shown to co-ordinately regulate transcription of mitotic genes (**Georlette et al., 2007**). Furthermore, we found that the phenotype of the recurrently mutated transcriptional repressor *spen* (SPEN) is aggravated by *nonC*, which is involved in nonsense-mediated mRNA decay (**Figure 7D**) (**Oswald et al., 2005**).

We detected complex epistatic network structures. For example, the transcriptional co-regulators *jumu* and *CtBP* and the nuclear pore complex component *Nup75* formed an epistatic network composed of an alleviating and two aggravating interactions (**Figure 7E**). The recurrently mutated DNA damage response gene *RecQ4* (WRN) aggravates *jumu*, while *Nup75* aggravates *Sin3A* (SIN3A/B). This results in a network structure containing a loop. Taken together, this demonstrates that our method of mapping epistatic interactions can predict network structures of regulatory activity.

## Discussion

The identification and mechanistic dissection of genetic interactions is a current frontier in molecular genetics. A systematic understanding of genetic interactions is an important basis for linking genotypes to phenotypes, to dissect the genetic dependencies of cellular processes and may also have profound implications for disease treatment. Identifying genetic dependencies can facilitate finding combinatorial drug treatments and avoiding resistance mechanisms, which are important in cancer (**Nijman and Friend, 2013**). The identification of genetic interactions in population-based observational studies, however, is limited by sample size and statistical power (**Zuk et al., 2012**) both in model organisms (**Mackay, 2014**) and for human diseases (**Civelek and Lusis, 2014**). Even though algorithmic and statistical approaches have been tuned over the last 10 years to address this issue, the number of detected instances of epistasis in genome-wide association studies remains small (**Lincoln et al., 2005**; **Gregersen et al., 2006**; **Wei et al., 2014**). Large-scale reverse-genetic combinatorial perturbation experiments offer an avenue to systematically map genetic interactions to infer gene functions and their interdependencies.

Analyses of genetic interaction screens so far have been mainly based on the 'statistical' definition by **Fisher (1918)**. A statistical genetic interaction exists between genes if the effect of their combined perturbation differs from what is expected from the single perturbations. The obtained information on gene–gene interactions is symmetric and does not reveal a potential directionality. The experimental and computational approach presented in this study aimed at capturing interactions, first irrespective of direction (symmetric), and second, adding directionality. Directional genetic interactions provide an additional layer of information above symmetric, statistical genetic interactions. Directional genetic interactions reflect causal, temporal or regulatory relationships between phenotypes and their

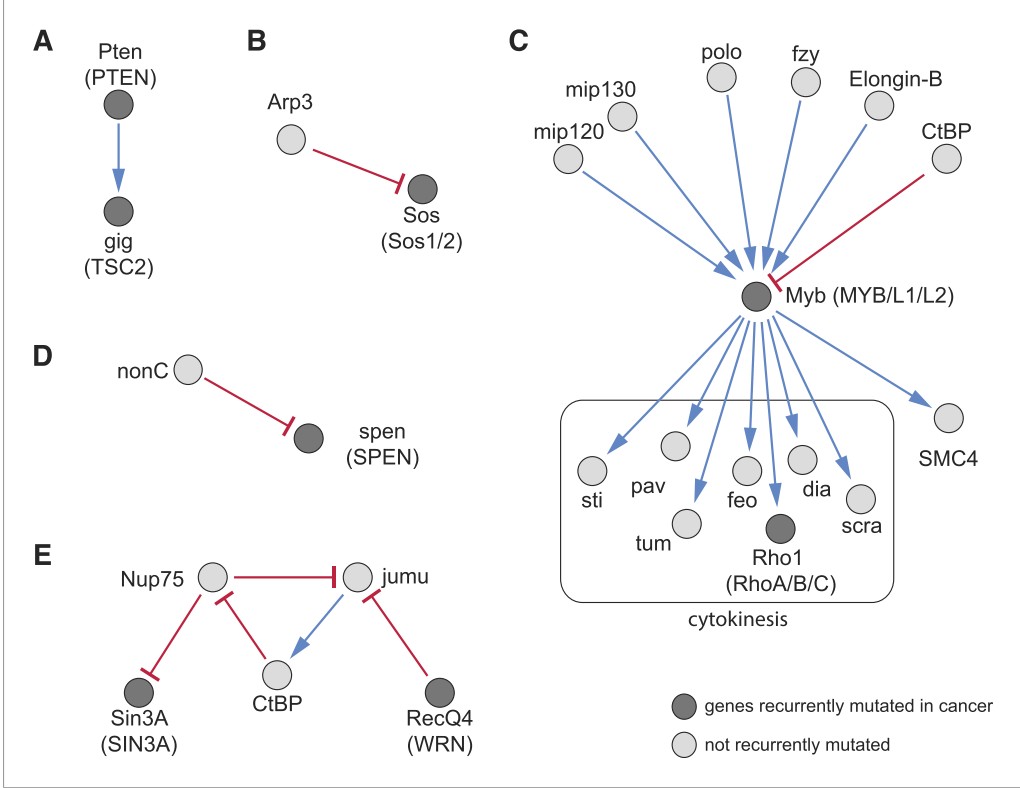

**Figure 7**. Recurrently mutated genes. (**A**–**E**) Subnetworks showing detected directional genetic interactions of recurrently mutated cancer genes. Recurrently mutated genes as reported by *Tamborero et al. (2013)* are coloured dark. Red arrows depict aggravating interactions and blue arrows depict alleviating interactions.

underlying biological processes. Our method requires the observation of multiple phenotypes and is an algorithmic formalisation of the epistasis concept of *Bateson (1907)*. Bateson mapped epistatic alleles when studying the inheritance of coat colour of mice using multiple colour phenotypes. The formalisation of mapping epistatic interactions as done in our study is straightforward to implement and should also be applicable to future studies.

Our map encompasses approximately 1300 genes for which we derived genetic interaction profiles, the largest genetic interaction map generated for a metazoan cell to date. The study demonstrates the feasibility of large-scale, automated discovery of directional epistatic interactions. There are also other approaches to annotate relationships between genes. For instance, *Kelley and Ideker (2005)* detected different interaction types by searching for network motifs and integrated information from genetic, physical and reaction networks. *Drees et al. (2005)* and *St Onge et al. (2007)* derived partial orderings of single and double mutant viability phenotypes. This method can derive directional interactions in special cases, for example, when one of the genes does not have a phenotype itself. More recently, *Vinayagam et al. (2014)* annotated protein–protein interactions with positive and negative signs, which they obtained from the correlations between the genes' individual knockdown phenotypes. *Liberali et al. (2014)* tested whether one gene's single knockdown phenotypes are contained in those of another gene, and used such subset relationships to derive hierarchies between genes. In contrast to these previous approaches, our study directly captures epistatic relationships from multivariate phenotypes observed after di-genic perturbations, often revealing effects that cannot be constructed from single-gene perturbations, or from single phenotypes alone. *van Wageningen et al. (2010)* recorded gene expression profiles from single and double yeast deletion strains and reported frequent instances of mixed epistasis. It would be interesting to study how our approach can be extended to detect mixed directional relationships. Another avenue for future work is the combination of directional genetic interaction mapping with Bayesian modelling (*Battle et al., 2010*) to further improve the inference of regulatory networks.

For the gene set presented here, we predicted the temporal and causal relationships between cell cycle regulators during mitosis without using prior knowledge. To highlight the functional relevance of the genetic interactions revealed in the study, we first showed that Ras-mediated signalling in fly development in vivo can be suppressed by loss-of function mutations in multiple SWI/SNF components. This is concordant with results from *Herr et al. (2010)*, who reported attenuated EGFR/Ras/MAPK signalling activity after reduced *brm* activity in the background of the dominant active EGFR mutation *ElpB1*. In addition, we confirmed the genetic interactions of SWI/SNF components with constitutively active KRAS in human colon cancer cells. Recent cancer genome sequencing studies have revealed many recurrent mutations that are individually rare (*Chin et al., 2011*). A current challenge is to order these genetic variants by their functional and phenotypic effects. We have demonstrated a method to detect which recurrently mutated genes modify each other's phenotypes. Such information could be instrumental for understanding the joint effect of multiple mutated genes, for example, in the same tumour, and may contribute to understanding tumour evolution and drug sensitivity.

We have demonstrated inference of directional genetic interactions from multiple phenotypes as a new approach to shed light on the patterns underlying complex genotype-phenotype relationships. This approach could provide a basis for the understanding of epistatic and functional dependencies between disease-associated genetic variants. Future studies could include additional phenotypes, optimized experimental designs and improved computational methods for the discovery of directional interactions.

## Materials and methods

### *Drosophila* tissue culture and combinatorial RNAi

Schneider S2 cells (D.Mel-2; ThermoFisher, Waltham, MA) were grown in serum-free medium (Express Five SFM; ThermoFisher) supplemented with 20 mM GlutaMAX (ThermoFisher) and 1% penicillin-streptomycin (ThermoFisher) as described previously (*Horn et al., 2011*). Two dsRNAs of length 200–225 nt and non-overlapping in sequence were designed for each target gene using NEXT-RNAi software (*Horn et al., 2010*) and annotation from FlyBase release r5.24 (*Supplementary file 1*). The algorithm was designed to avoid perfect matches of length $\geq 19$ nt or overall sequence similarity (BLAST E-value $>10^{-10}$) with non-target transcripts, more than six tandem tri-nucleotide repeats of the type CAN, low complexity regions and UTRs. The library was synthesized in 96-well plates by two-step PCR approach followed by in vitro transcription (*Steinbrink and Boutros, 2008*). dsRNA products were cleaned using filter plates (MultiScreen HTS PCR 96-Well, Millipore) with columns of Bio-Gel P-30 (Bio-Rad). Concentrations were measured using a NanoDrop 8000 spectrophotometer (Thermo Scientific) and individually adjusted for each dsRNA using a Beckman Coulter Biomek FX Span-8.

For combinatorial RNAi, a template-query experimental design was used as previously described (*Horn et al., 2011*). 100 ng (2 µl) of each template (target) dsRNA were pipetted into 384-well clear-bottom microscopy plates (BD Falcon). 100 ng (2 µl) of one of the query dsRNAs were added to all wells of the plates except for column 2 (which contained control reagents) using a NanoDrop II dispenser (Innovadyne). In each well, 6500 cells were seeded in 30 µl culture medium with 0.15 µl of 0.4 mg ml$^{-1}$ DDAB per well and incubated for 5 days at 25°C before fixation and staining.

### Automated microscopy

Cells were stained using a Beckman Coulter Biomek FX liquid handling workstation with 384-well tip head. Of the 40 µl assay volume, 15 µl were removed, and cells were fixed and permeabilised for 60 min at room temperature by addition of 40 µl of 6% PFA (Sigma), 0.3% TX-100 (AppliChem) solution in PBS (Sigma). Cells were washed by removing 40 µl supernatant and adding 50 µl PBS. 50 µl of the supernatant were removed and 10 µl of 4% BSA (GERBU), 0.1% TX-100 (in PBS) containing monoclonal phospho-Histone H3 antibody (conjugated to Alexa Fluor 647, Cell Signaling, #3458BC) and monoclonal anti-α-Tubulin antibody (conjugated to FITC, Sigma F2168), both diluted 1:750, were added and incubated overnight at 8°C (protected from light). The next day 10 µl supernatant were removed, and DNA was stained by addition of 40 µl Hoechst 33,342 (ThermoFisher) diluted 1:2000 in PBS and incubation for 30 min at room temperature. Finally, cells were washed once by removing 40 µl supernatant and adding 50 µl PBS and washed twice more by removing 50 µl supernatant and adding 50 µl PBS. Plates were sealed with aluminium sealing tape (Corning) and imaged directly or stored at

8°C until imaging. Fluorescence images were acquired on an INCell Analyzer 2000 (GE Healthcare). Whole-well, two-channel images were acquired using the 4× objective (Hoechst/DNA and Alexa647/phospho-Histone H3). Three-channel images (additionally including FITC/α-Tubulin) of higher resolution were acquired using the 10× objective.

## Image segmentation and feature extraction

Image analysis was adapted from previously described methods (*Carpenter et al., 2006*; *Fuchs et al., 2010*; *Held et al., 2010*; *Pau et al., 2010*), using the R package EBImage (*Pau et al., 2010*). Nuclei were segmented by adaptive thresholding, identified separately in the DAPI and pH3 channels and matched. From the 4× images, 84 features per nucleus were extracted, including number of nuclei in each channel, mean, standard deviation and other distribution statistics of area and fluorescence intensity in both channels. Local cell density was estimated with a kernel density estimator using a Gaussian kernel on eight different scales. In the 10× images, the cellular boundary was estimated from fluorescence levels in the α-tubulin channel by extension from nuclei by the propagation algorithm (*Jones et al., 2005*). 244 features per cell were extracted from the 10× images, including distribution moments, shape and Haralick texture features (*Haralick et al., 1973*). To obtain features per experiment (cell population), averages of per-cell features were computed separately for mitotic and non-mitotic cells. This yielded 328 features, including number of nuclei, local cell density, mean, standard deviation and other distribution statistics of area and fluorescence intensity in all channels. A full list of features is provided in *Supplementary file 3*. All feature data were subjected to a variance-stabilising transformation (*Huber et al., 2002*) that interpolates between the logarithm function for large values and a linear function around 0. The transformation reduced heteroskedasticity and improved the fit of the multiplicative model, which was verified by the concentration of residuals around 0.

## Feature interpretation and selection

Many of the extracted features have obvious or intuitive interpretations, such as mean cell area, eccentricity, intensity. Our measurements for these phenotypes correlate with previously published single gene morphological phenotypes (*Rohn et al., 2011*) for cell size and eccentricity (*Figure 1—figure supplement 3*). Other features are more abstract. For instance, we included the quantile features to be able to detect changes that affect only a small proportion of cells: for example, the elongation phenotype often only affects a small fraction of the cells. Other features are mathematical descriptors, or numerical 'fingerprints', of texture and intensity patterns; the use of these is well-established in cellular image analysis (*Boland and Murphy, 2001*), and while they are not directly interpreted as a specific process or function, they sensitively pick up complex changes in the cellular network caused by the genetic perturbations. Together, we computed a large number of phenotypic features, which were partially redundant. We then aimed at selecting an informative, non-redundant subset of these features for further analysis by an automated procedure. First, we eliminated those features that were not sufficiently reproducible between replicates (cor <0.6) (*Figure 1C*). We then started the feature selection with three manually chosen features based on their interpretability and to facilitate comparability with previous work: number of cells, fraction of mitotic cells and cell area (*Horn et al., 2011*). Subsequently, for each further feature, we fit a linear regression that modelled the feature's values over all experiments as a function of the already selected features. The residuals of this fit were used to measure new information content not yet covered by the already selected features (*Figure 1D*). Among candidate features, we selected the one with the maximum correlation coefficient of residuals between two replicates. This procedure was iterated, and in each iteration we computed the fraction of features with positive correlation coefficients. The iteration was stopped when this fraction no longer exceeded 0.5 (*Figure 1D*). This choice of cut-off was motivated by the fact that for a set of uninformative features, the fraction is expected to be 0.5. The 21 selected features are listed in *Supplementary file 4*.

## Pairwise interaction scores

Pairwise interaction scores (π-scores) were computed for each pair of dsRNA reagents and each of the 21 selected features as described before (*Horn et al., 2011*). To summarize the four measurements for each gene pair, they were tested against the null hypothesis that their mean was zero. p-values were computed by the moderated *t*-test implemented in the R package *limma* (*Smyth, 2004*) and adjusted for multiple testing by the method of Benjamini-Hochberg (*Benjamini and Hochberg, 1995*).

*Figure 2A* shows the number of interactions (positive and negative interactions) for each feature and the cumulative number of interactions.

## Directed genetic interactions

For all gene pairs with a significant interaction (FDR 0.01) in at least one phenotype, we aimed to compute the direction of phenotypic regulation. For each gene pair the π-score-vector over all 21 features was fit by a linear combination of the main effect vectors of the two genes. By analysis of variance we then quantified how much of the π-score was explained by each of the two genes. We called an epistatic interaction from gene A to B if the fraction of sum of squares explained by gene A was smaller than a lower threshold and if the fraction of sum of squares explained by gene B was larger than a higher threshold. The two thresholds were chosen as the 10% and 95% quantile over all gene pairs. In addition to the direction of phenotypic regulation, the coefficient of the fit defined the sign, which could be alleviating (negative coefficient) or aggravating (positive coefficient).

## Genetic interactions of SWI/SNF member loss-of-function alleles with $Ras85D^{V12}$ allele in vivo

The following *Drosophila* stocks were obtained from the Bloomington stock center: $osa^2/TM6B$, $Tb^1$ (#3616); $P(PZ)Snr1^{01319}$ $ry^{506}/TM3$, $ry^{RK}$ $Sb^1$ $Ser^1$ (#11529) and $brm^2$ $e^s$ $ca^1/TM6B$, $Sb^1$ $Tb^1$ $ca^1$ (#3619). These flies were crossed to $Q^1/CyO$, $P\{ry[+t7.2] = sevRas85D^{V12}\}FK1$ (also called Q1/CyO, P(sev-Ras1$^{V12}$) FK1, Bloomington stock #3795). Progeny containing $CyO$, $P\{ry[+t7.2] = sevRas85D^{V12}\}FK1$ and either heterozygous mutant candidate genes or wild-type ($w^{1118}$) background were scored for rough eye phenotypes. Crosses were reared at 25°C and phenotypes were scored independently by two experienced researchers.

## Scanning electron microscopy (SEM)

*Flies containing CyO, P{ry[+t7.2] = sevRas85D$^{V12}$}FK1* and either heterozygous mutant candidate genes or hetrogyzous wild-type ($w^{1118}$) background were selected. Since all experimental flies were red-eyed thus red-eyed Oregon-R (+/+) flies were used as control. *Drosophila* heads were prepared for SEM by serial dehydration in 20%, 50%, 70%, 90%, 100% ethanol, 30 min each step. Heads were further dehydrated with critical point dehydration using Leica Critical Point Dryer CPD300. Heads were then treated with 15 nm Gold particle sputtering using Leica EM MED020. Scanning electron images were acquired with Zeiss Leo1530 SEM at 3 KV and WD = 5 mm.

**Table 1**. siRNAs

| Gene | Target sequence | Cat # |
|---|---|---|
| CTRL/control | SIGENOME NON TARGET POOLNO2 | D-00120614-20 |
| ARID1A | GCAACGACAUGAUUCCUAU | MU-017263-01 (siGENOME) D-017263-01 |
| ARID1A | GAAUAGGGCCUGAGGGAAA | MU-017263-01 (siGENOME) D-017263-02 |
| ARID1A | AGAUGUGGGUGGACCGUUA | MU-017263-01 (siGENOME) D-017263-03 |
| ARID1A | UAGUAUGGCUGGCAUGAUC | MU-017263-01 (siGENOME) D-017263-04 |
| SMARCA4 | GAAAGGAGCUGCCCGAGUA | MU-010431-00 (siGENOME) D-010431-01 |
| SMARCA4 | CCAAGGAUUUCAAGGAAUA | MU-010431-00 (siGENOME) D-010431-02 |
| SMARCA4 | GAAAGUGGCUCAGAAGAAG | MU-010431-00 (siGENOME) D-010431-03 |
| SMARCA4 | AGACAGCCCUCAAUGCUAA | MU-010431-00 (siGENOME) D-010431-04 |
| SMARCB1 | GAAACUACCUCCGUAUGUU | MU-010536-01 (siGENOME) D-010536-01 |
| SMARCB1 | CCACAACCAUCAACAGGAA | MU-010536-01 (siGENOME) D-010536-02 |
| SMARCB1 | GUGACGAUCUGGAUUUGAA | MU-010536-01 (siGENOME) D-010536-03 |
| SMARCB1 | AGACCUACGCCUUCAGCGA | MU-010536-01 (siGENOME) D-010536-04 |

## Cell viability assay

HCT116 PAR007 and HCT116 HD104-008 (KRAS mtG12 KO) cell lines were ordered from HORIZON. Cells were cultured in McCoy's medium (Gibco) supplemented with 10% FCS and penicillin-streptomycin at 37˚C in 5% $CO_2$. Cells were transfected in a 384-well format using Lipofectamine RNAiMAX (ThermoFisher) diluted in serum-free McCoy's medium. Genes were targeted using a pool of four siRNAs (Thermo Fisher Scientific) with a total final concentration of 50 µM. Cell viability was measured using CellTiter-Glo Luminescent Viability Assay (Promega) using a Mithras LB 940 Multimode Microplate Reader (Berthold). Each biological replicate was performed in technical quadruplicates. The experiment was conducted in four independent biological replicates (*Table 1*).

## Data and source code

A documented software package is available as the R/Bioconductor package DmelSGI. (http://bioconductor.org) The package contains all software that is used for analysis and contains a vignette with the documented source code. The code vignette is provided as *Source code 1*.

## Acknowledgements

We thank Nassos Typas, Sarah Ayling, Marija Buljan, Marco Breinig and Felix Klein for helpful comments on the manuscript, and members of the Boutros and Huber groups for critical discussions. We thank the CellNetworks EMCF facility for help with electron microscopy. BF and TS were supported by the CellNetworks Cluster of Excellence of the German Research Foundation (DFG). Research in the lab of MB is supported by an ERC Advanced Grant ("Syngene") of the European Research Council. Equipment funding has been provided by the DFG. WH acknowledges support by the EU project Systems Microscopy.

## Additional information

### Funding

| Funder | Grant reference | Author |
|---|---|---|
| European Research Council (ERC) | | Michael Boutros |
| European Commission | Systems Microscopy Network of Excellence | Wolfgang Huber |
| Deutsche Forschungsgemeinschaft (DFG) | | Wolfgang Huber |
| Deutsche Forschungsgemeinschaft (DFG) | CellNetworks Cluster of Excellence | Bernd Fischer, Thomas Sandmann |

The funders had no role in study design, data collection and interpretation, or the decision to submit the work for publication.

### Author contributions

BF, TS, TH, Conception and design, Acquisition of data, Analysis and interpretation of data, Drafting or revising the article; MB, Acquisition of data, Analysis and interpretation of data, Drafting or revising the article; VC, Acquisition of data, Analysis and interpretation of data; WH, MB, Conception and design, Analysis and interpretation of data, Drafting or revising the article

### Author ORCIDs

Bernd Fischer, http://orcid.org/0000-0001-9437-2099

## Additional files

### Supplementary files

• Supplementary file 1. A table of primer and dsRNA sequences for all targeted genes.

- Supplementary file 2. A table of query genes.

- Supplementary file 3. A table of extracted features.

- Supplementary file 4. A table of selected non-redundant features.

- Supplementary file 5. A table of genes passed quality control.

- Supplementary file 6. A table of directed genetic interactions.

- Source code 1. Documented source code for reproducing all figures and tables.

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
