## [Decision Letter]

Thank you for sending your work entitled “A Map of Directional Genetic Interactions in a Metazoan Cell” for consideration at *eLife*. Your article has been favorably evaluated by Fiona Watt (Senior editor) and 3 reviewers, one of whom is a member of our Board of Reviewing Editors, and one of whom, Trey Ideker, has agreed to share his identity.

The Reviewing editor and the other reviewers discussed their comments before we reached this decision, and the Reviewing editor has assembled the following comments to help you prepare a revised submission.

The manuscript by Boutros and colleagues reports analysis of a large-scale screen evaluating effects of siRNA knockdown of genes alone and in pairwise combinations on multiple quantitative cellular phenotypes. The major novel innovation of the manuscript is the use of combinations of phenotypes as the basis for determining the directionality of epistatic interactions. The authors present evidence for directional epistatic relationships between genes encoding components of the Swi/Snf complex and genes influencing Ras signaling as an unexpected and significant finding. The authors present evidence for this functional relationship in the fly eye and in a human colon cancer cell line. All three reviewers liked the basic concept and execution of the experiments and thought that the manuscript would be a good candidate for *eLife* subject to satisfactory responses to the following major comments:

1) The Introduction needs to be revised to give additional consideration to prior work. There was some disagreement among reviewers regarding the extent to which earlier work communicates directional interactions. However, at a minimum the following additional references should be cited and referred to in the Introduction and/or Discussion: St. Onge et al. Nature Genetics (2007); Drees et al. Genome Biology (2005); Kelley et al. Nature Biotech (2005). See also Jerby-Arnon et al. Cell 2014 with respect to use of the term 'statistical' genetic interactions.

2) The Introduction should be made more accessible to the general reader by discussing the mechanistic origins of interactions at the protein level.

3) There was unanimous agreement among all three reviewers that Figure 4 needs to be better explained.

4) For the single gene knockdown phenotypes, these should be benchmarked against previous RNAi knockdown high-content imaging papers to look at the reproducibility/overlap.

5) The biological meanings of many of the cellular phenotypes that are used as the basis for determining directionality of epigenetic interactions are unclear. While some phenotypes such as cell size, cell number, nuclear size, etc., can be traced to functions of specific genes or combinations of genes, many or most 'phenotypes' are hard to interpret (e.g., 10x.meanNonmitotic.cell.BDAPITub.b.q001, from Table S3). This raises the question of whether epistatic relationships observed for many of the quantitative parameters are of biological relevance. The authors do provide evidence that each of the quantitative phenotypes chosen for study add information and that their screen and analytical methods capture previously established functional relationships, but additional explanation of why many of the seemingly esoteric measurements should be informative at a biological level is needed.

6) There are prior reports of connections between Swi/Snf complexes and RAS signaling (e.g., PMID: 20416294, PMID: 23438604). In some cases these interactions would seem to predict opposite results from those observed in human colon cancer cells. These studies possibly weaken the interpretation and novelty of what the authors have chosen to highlight as a new and unexpected finding. The authors need to place their findings in the context of these prior papers. It would also be helpful to know what phenotypes besides cell number drove the epistatic relationships and their directionality for the plot shown in Figure 6.

---

## [Author Response]

*1) The Introduction needs to be revised to give additional consideration to prior work. There was some disagreement among reviewers regarding the extent to which earlier work communicates directional interactions. However, at a minimum the following additional references should be cited and referred to in the Introduction and/or Discussion: St. Onge et al. Nature Genetics (2007); Drees et al. Genome Biology (2005); Kelley et al. Nature Biotech (2005)*.

We thank the reviewers for highlighting these studies. In the revised version of the manuscript, we acknowledge these papers and refer to them either in the Introduction (second paragraph) or Discussion (third paragraph). We also explain how our approach differs from previous studies. Specifically, in previous studies epistasis and therefore directionality was only inferred in special cases, e.g. where one of the single gene phenotypes is zero. By using multiple phenotypes and considering them as ‘vectors’, our method is more general and more widely applicable. Figure 3 gives an example: when both genes have some effect on growth rate, it is not possible to unambiguously derive directionality. In practice, these cases are common and previous studies have demonstrated that genes with phenotypes are most likely to show genetic interactions (Costanzo, Science, 2010). In contrast, our multivariate method is able to assign directionality even in cases where both single gene phenotypes are non-zero, based on a straightforward vector algebra calculation.

*See also Jerby-Arnon et al. Cell 2014 with respect to use of the term 'statistical' genetic interactions*.

Regarding the nomenclature of epistasis and genetic interactions, a range of different definitions is found in the literature. In particular, the term “statistical” was used by Jerby-Arnon et al. to detect and prioritize synthetic lethal or synthetic dosage lethal interactions by an integrative analysis of diverse experimental datasets by three separate statistical inference procedures. While they report the use of statistical methods to discover genetic interactions, in the paper they did not use the expression “statistical genetic interactions” for these. To the best of our knowledge, we follow the mainstream usage of this term, where a statistical interaction is detected from an interaction term between the covariates in a linear model (ANOVA) for the phenotype (see e.g., Cordell et al., Epistasis: what it means, what it doesn't mean, and statistical methods to detect it in humans, Human Molecular Genetics, 2002). To clarify the terminology, we added an explanation in the Introduction (second paragraph).

*2) The Introduction should be made more accessible to the general reader by discussing the mechanistic origins of interactions at the protein level*.

We added several sentences on genetic interactions and their underlying mechanistic origins (parallel pathways; linear pathway or protein complex) to the Introduction (second paragraph). In addition, we added a reference to the review of Dixon et al. (Andrews and Boone labs) in the Annual Review of Genetics (2009), which gives a more extensive overview over possible mechanistic origins.

*3) There was unanimous agreement among all three reviewers that*
Figure 4
*needs to be better explained*.

We thank the reviewers for pointing this out. We have revised the figure to get a better connection between the bar charts in the first column and the geometric charts in the third column. We also revised and extended the text by adding additional explanations.

*4) For the single gene knockdown phenotypes, these should be benchmarked against previous RNAi knockdown high-content imaging papers to look at the reproducibility/overlap*.

We compared the extracted phenotypes to a previous screen in *Drosophila* S2R+ cells by Rohn et al. (J Cell Biol, 2011) and added Figure 1—figure supplement 3 to the manuscript, which shows a significant overlap of the phenotypes (fourth paragraph of Materials and Methods). However, we would like to point out the limitations of such an analysis as different cell lines have been used.

*5) The biological meanings of many of the cellular phenotypes that are used as the basis for determining directionality of epigenetic interactions are unclear. While some phenotypes such as cell size, cell number, nuclear size, etc., can be traced to functions of specific genes or combinations of genes, many or most 'phenotypes' are hard to interpret (e.g., 10x.meanNonmitotic.cell.BDAPITub.b.q001, from Table S3). This raises the question of whether epistatic relationships observed for many of the quantitative parameters are of biological relevance. The authors do provide evidence that each of the quantitative phenotypes chosen for study add information and that their screen and analytical methods capture previously established functional relationships, but additional explanation of why many of the seemingly esoteric measurements should be informative at a biological level is needed*.

We added the following explanation on this issue to Section “Feature interpretation and selection” in the Methods section.

“Many of the extracted features have obvious or intuitive interpretations, such as mean cell area, eccentricity, intensity. Our measurements for these phenotypes correlate with previously published single gene morphological phenotypes (51) for cell size and eccentricity (Figure 1—figure supplement 3). Other features are more abstract. For instance, we included the quantile features to be able to detect changes that affect only a small proportion of cells: e.g. the elongation phenotype often only affects a small fraction of the cells. Other features are mathematical descriptors, or numerical ‘fingerprints’, of texture and intensity patterns; the use of these is well-established in cellular image analysis (11), and while they are not directly interpreted as a specific process or function, they sensitively pick up complex changes in the cellular network caused by the genetic perturbations.”

Regarding the biological relevance, we can only point out that the detection of apparent changes in the output of a biological system in order to sense variation in the underlying, complex molecular network, often in a very specific manner, has a long history. For instance, medical practitioners can sometimes diagnose congenital anomalies from very specific, recurrent symptoms (syndromes), even if we currently cannot make an intuitive biology-based link. Differences between vein pattern formations in *Drosophila* have been used to map intra- and inter-cellular signalling networks of very general relevance. Similarly, we do not attempt to link interactions found with respect to one of these more abstract phenotypes in our data directly to a biological function. We merely use them to detect the relationships between the underlying perturbations.

*6) There are prior reports of connections between Swi/Snf complexes and RAS signaling (e.g., PMID: 20416294, PMID: 23438604). In some cases these interactions would seem to predict opposite results from those observed in human colon cancer cells. These studies possibly weaken the interpretation and novelty of what the authors have chosen to highlight as a new and unexpected finding. The authors need to place their findings in the context of these prior papers. It would also be helpful to know what phenotypes besides cell number drove the epistatic relationships and their directionality for the plot shown in*
Figure 6.

Herr et al. (PMID: 20416294) reported an antagonistic effect of geminin (gem) and the SWI/SNF component brahma (brm) on EGFR-Ras-MAPK signaling in *Drosophila*. They showed that while *brm* is required for EGFR-Ras-MAPK signaling, *gem* inhibits downstream activity. Their interaction phenotype of the over-activated EGFR pathway with *brm* heterozygous mutant is in agreement with our data. However, this study did not show an effect of other SWI/SNF members *osa* and *Snr1* on EGFR pathway. This is now discussed in the fourth paragraph of the Discussion. Tu et al. (PMID: 23438604) reported the requirement of BRG1 (the human brm ortholog) in Ras-induced senescence. They use a lung-derived fibroblast cell line. While loss of BRCA1 is known to induce senescence, they show that the effect of BRG1 is independent of BRCA1 or DNA damage. While the consequences of activated Ras signaling induce different cell fate routes, we demonstrate a conserved dependency of the increased growth phenotype on SWI/SNF function.

It is indeed interesting to look at the features that drive the epistatic relationship. Multiple features contribute, but most of them are related to nuclear size and eccentricity. We added a panel (Figure 6) that exemplary shows these relationships between RasGap1 and the SWI/SNF members. It can be observed that the RasGap1 growth rate increase is compensated after double knockdown, but the SWI/SNF cell area and eccentricity phenotype is strongly increased after double knockdown.